# Unsupervised Neural Multi-document Abstractive Summarization of Reviews

## Abstract

Abstractive summarization has been studied using neural sequence transduction methods with datasets of large, paired document-summary examples. However, such datasets are rare and the models trained from them do not generalize to other domains. Recently, some progress has been made in learning sequence-to-sequence mappings with only unpaired examples. In our work, we consider the setting where there are only documents (product or business reviews) with no summaries provided, and propose an end-to-end, neural model architecture to perform unsupervised abstractive summarization. Our proposed model consists of an auto-encoder trained so that the mean of the representations of the input reviews decodes to a reasonable summary-review. We consider variants of the proposed architecture and perform an ablation study to show the importance of specific components. We show through metrics and human evaluation that the generated summaries are highly abstractive, fluent, relevant, and representative of the average sentiment of the input reviews.

## 1 Introduction

Supervised, neural sequence-transduction models have seen wide success in many language-related tasks such as translation (Wu et al., 2016; Vaswani et al., 2017) and speech-recognition (Chiu et al., 2017). In these two cases, the model training is typically focused on the translation of sentences or recognition of short utterances, for which there is an abundance of parallel data. The application of such models to longer sequences (multi-sentence documents or long audio) works reasonably well in production systems because the sequences can be naturally decomposed into the shorter ones the models are trained on and thus sequence-transduction can be done piece-meal.

Similar neural models have also been applied to abstractive summarization, where large numbers of document-summary pairs are used to generate news headlines (Rush et al., 2015) or bullet-points (Nallapati et al., 2016; See et al., 2017). Work in this vein has been extended by Liu et al. (2018) to the multi-document[1] case to produce Wikipedia article text from references documents.

However, unlike translation or speech recognition, adapting such summarization models to different types of documents without re-training is much less reasonable; for example, in general documents do not decompose into parts that look like news articles, nor can we expect our idea of saliency or desired writing style to correspond with that of particular news publishers. Re-training or at least fine-tuning such models on many in-domain document-summary pairs should be expected to get desirable performance. Unfortunately, it is very expensive to create a large parallel summarization corpus and the most common case in our experience is that we have many documents to summarize, but have few or no examples of summaries.

We side-step these difficulties by completely avoiding the need for example summaries. Although there has been previous work on extractive summarization without supervision, we describe, to our knowledge, the first end-to-end, neural-abstractive, unsupervised summarization model. Unlike recent approaches to unsupervised translation (Artetxe et al., 2017; Lample et al., 2017), we do not only assume there is no parallel data, but also assume no dataset of output sequences.

In this paper, we study the problem of abstractively summarizing multiple reviews about a business or product without any examples (in fact they do not exist in our dataset) and apply our method

---

[1]Note multi-document here means multiple documents about the same topic.

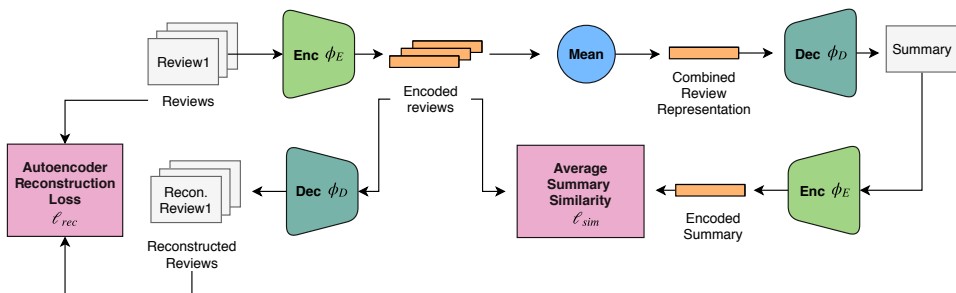

Figure 1: The proposed model architecture.

to publically available Yelp[2] and Amazon reviews (McAuley et al., 2015). We describe an architecture for summarizing multiple reviews in the form of a single review, perform multiple ablation experiments to justify the architecture chosen, and define proxy metrics to evaluate our generated summaries in the absence of ground-truth. Through these metrics, crowd-sourced human evaluation, and qualitative analysis, we show that the generated summaries are often fluent, relevant, and representative of the summarized reviews.

## 2 MODELS AND METHODS

The model consists of two main components: (1) an auto-encoder module that learns representations for each review and constrains the generated summaries to be in the language domain, and (2) a summarization module that learns to generate summaries that are semantically similar to each of the input documents. These contribute a reconstruction loss and similarity loss, respectively. Both components contain an LSTM encoder and decoder – the two encoders' weights are tied, and the two decoders' weights are tied. The encoder and decoder are also initialized with the same pre-trained language model trained on the reviews of the dataset. The overall architecture is shown in Figure 1.

Suppose we have an invertible tokenizer, $T$, that maps text documents, $\mathbb{D}$, to sequences of tokens (from a fixed vocabulary), $T(\mathbb{D})$. Let $\mathbb{V} \subset T(\mathbb{D})$ represent the tokenized reviews in our dataset with a maximum length of $L$. Given a set of $k$ reviews about an entity (business or product), $\{x_1, x_2, ..., x_k\} \subset \mathbb{V}$, we would like to produce a document tokenized using the same vocabulary, $s \in T(\mathbb{D})$, that summarizes them.

In the auto-encoder sub-module, an encoder $\phi_E : \mathbb{V} \mapsto \mathbb{R}^n$, maps reviews to real-vector codes, $z_j = \phi_E(x_j)$. $\phi_E(x) = [h, c]$ is implemented as the concatenation of the final hidden and cell states of an LSTM (Hochreiter & Schmidhuber, 1997) after processing $x$ one token at a time. A second decoder LSTM defines a distribution over $\mathbb{V}$ conditioned on the latent code, $p(x|z_j) = \phi_D(z_j)$, by initializing its state with $z_j$, and is trained using teacher-forcing (Williams & Zipser, 1989) with a standard cross-entropy loss to reconstruct the original reviews, i.e. the auto-encoder is implemented as a sequence-to-sequence model (Sutskever et al., 2014).

$$\ell_{rec}(\{x_1, x_2, ..., x_k\}, \phi_E, \phi_D) = \sum_{j=1}^{k} \ell_{cross\_entropy}(x_j, \phi_D(\phi_E(x_j))) \qquad (1)$$

In the summarization module, $\{z_1, z_2, ..., z_k\}$ are combined using a simple mean over the hidden and cell states, $\bar{z} = [\bar{h}, \bar{c}]$, which is decoded by $\phi_D$ into the summary $s$. By using the same decoder as the auto-encoder, $\phi_D$, we constrain the output summary to the space of reviews, $s \in \mathbb{V}$, and can think of it as a *canonical review*. We then re-encode the summary and compute a similarity loss that further constrains the summary to be semantically similar to the original reviews; we use average cosine distance, $d_{cos}$, between the hidden states $h_j$ of each encoded review and $h_s$ of the encoded

[2]https://www.yelp.com/dataset/challenge

summary, $\phi_E(s) = [h_s, c_s]$.

$$s \sim \phi_D(\bar{z}) \tag{2}$$

$$\ell_{sim}(\{x_1, x_2, ..., x_k\}, \phi_E, \phi_D) = \frac{1}{k} \sum_{j=1}^{k} d_{cos}(h_j, h_s) \tag{3}$$

As we lack ground truth summaries, we cannot use teacher forcing to generate the summary in Equation (2). Instead, we generate the summary using the Straight Through Gumbel-Softmax trick (Jang et al., 2016; Maddison et al., 2016), which approximates sampling from a categorical distribution (in this case a softmax over the vocabulary) and allows gradients to be backpropagated through this discrete generation process. We note that this sampling procedure allows us to avoid the exposure bias (Ranzato et al., 2015) of teacher-forcing, as the summary is generated through the same procedure during training and as in inference.

The final loss we optimize is simply $\ell_{model} = \ell_{rec} + \ell_{sim}$. We explored non-equal weighting of the losses but did not find a meaningful difference in outcomes.

## 2.1 METRICS AND EVALUATION

Though we have no ground truth summaries to compute traditional summarization metrics, we calculate three automatic statistics to guide model development and conduct a final human evaluation of summary quality.

**Rating accuracy**. As reviews are used by consumers to guide purchasing decisions, a useful summary should reflect the overall sentiment of the reviews. We first separately train a CNN-based classification model that given a review $x$, predicts the star rating of a review, an integer from 1 to 5. For each summary, we check whether the classifier's max predicted rating is equal to the average rating of the reviews being summarized (rounded to the nearest star rating). This binary accuracy is averaged across all the data points.

$$\text{Rating accuracy} = \frac{1}{N} \sum_{i=1}^{N} \left[ \mathbf{CLF}(s^{(i)}) == round\left(\frac{1}{k} \sum_{j=1}^{k} r_j^{(i)}\right) \right] \tag{4}$$

where $N$ is the number of data points, $\mathbf{CLF}$ is the trained classifier, $\mathbf{CLF}(s^{(i)})$ is the rating with the highest predicted probability, $s^{(i)}$ is the summary for the $i$-th data point, and $r_j^{(i)}$ is the rating for the $j$-th review in the $i$-th data point.

**Word Overlap (WO) score**. It is possible that a summary has the appropriate sentiment but is not grounded in information found in the reviews. As a sanity check that the summary is on-topic, we compute a measure of word overlap using the ROUGE-1 score (Lin, 2004) between the summary and each review and then average these scores, as shown in Equation 5. ROUGE is typically used between a candidate summary and a reference summary (which we lack), but its use here similarly captures how much the candidate summary encapsulates the original documents. We note that in our use case, this metric is highly biased towards extractive summaries, as the "reference" is the original reviews themselves and maximizing it is not necessarily appropriate; however, very little word overlap is likely pathological.

$$\text{Word Overlap score} = \frac{1}{N} \sum_{i=1}^{N} \left[ \frac{1}{k} \sum_{j=1}^{k} \mathbf{ROUGE}(s^{(i)}, r_j^{(i)}) \right] \tag{5}$$

**Negative Log-Likelihood (NLL)**. Generated summaries should also be fluent language. To measure this, we compute the negative log-likelihood of the summary according to a language model trained on the reviews. This metric is used to compare the outputs from different variations of our abstractive model.

**Human Evaluation**. To further assess the summaries and the validity of our metrics, we ran Mechanical Turk experiments (more details in Appendix B) asking workers to rate 100 summaries on a scale of 1 (very poor) to 5 (very good):

1. how well the sentiment of the summary agrees with the overall sentiment of the original review;
2. how well information is summarized across reviews;
3. the fluency of the summary based on five dimensions previously used in DUC-2005 Dang (2005): Grammaticality, Non-redundancy, Referential clarity, Focus, and Structure and Coherence.

## 2.2 BASELINE MODELS

**No training.** Perhaps the language model alone would be sufficient for generating summaries. This variant has the same architecture and language model initialization as our model, but we do not optimize $\ell_{model}$. The intent of this baseline is to show whether optimizing $\ell_{model}$ improves the quality of summaries beyond pre-training. As in the full model, the summary is generated by encoding the original reviews with $\phi_E$, computing the combined review representation, and decoding the summary using $\phi_D$.

**Extractive**. We use a recent, near state-of-the-art, centroid-based multi-document summarization method that uses word embeddings (Mikolov et al., 2013) instead of TF-IDF to represent each sentence (Rossiello et al., 2017). The maximum length of summaries was set to the $99.5^{th}$ percentile of reviews less than length $L$. This was chosen after evaluating various ceilings (e.g. $75^{th}$ percentile, $90^{th}$ percentile, no ceiling) on the validation set.

**Best Review**. It could be the case that one of the reviews would be a good summary. We thus compute the WO scores using each review as a summary. The review with the highest average (not including itself) WO score is selected.

**Worst Review**. We use the same procedure as the Best Review baseline, except we select the review with the lowest average WO score to get an idea of what is a bad word overlap score.

**Multi-Lead-1**. The Lead-$m$ baseline is often a strong baseline in single document summarization tasks and consists of the first $m$ sentences in the document (See et al., 2017). We create an analog by first randomly shuffling the reviews, and then adding the first sentence from each review until the maximum length $L$ is reached. If $L$ is not reached, then the summary is composed of the first sentence from each review.

## 2.3 MODEL VARIATIONS

To investigate the importance of different components and features, we evaluated the following variations of our model:

**No pre-trained language model.** Instead of initializing the encoders and decoders with the weights of a pre-trained language model, the entire model was trained from scratch.

**No auto-encoder.** To test our belief that the auto-encoder is critical for (a) keeping the summaries in the review-language domain, $\mathbb{V}$, and (b) producing review representations that actually reflect the review, we tested a variant without the auto-encoder. This model is shown in Appendix A.

**Reconstruction cycle loss**. A perhaps more straightforward model architecture would be to encode the reviews, compute $\bar{z}$, generate the summary $s$, and then use $s$ to decode into the reconstructed reviews $\hat{x}^j$, which would be used in a reconstruction loss with the original reviews. This last step would enforce the same constraint as the auto-encoding loss and the cosine similarity loss. This model is shown in Appendix A.

**Early cosine loss**. In the similarity loss, instead of computing distance between encoded reviews and the encoded summary, $\phi_E(s)$, we use $\bar{z}$:

$$\ell_{sim}(\{x_1, x_2, ..., x_k\}, \phi_E, \phi_D) = \frac{1}{k}\sum_{j=1}^{k} d_{cos}(z_j, \bar{z}) \tag{6}$$

Perhaps this alone would be enough to push $\bar{z}$ into a latent space suitable for decoding into a summary. This would also preclude the need for back-propagating gradients through the discrete sampling step – we only need to decode the summary at test time, which we do through greedy decoding. In contrast to our model, summary generation here suffers from the exposure bias of teacher-forcing. This model is shown in Appendix A.

**Untied decoders/encoders**. We relax the constraint that the review and summary decoders/encoders share weights.

## 3 RELATED WORK

Many popular extractive summarization techniques do not require example summaries and instead consider summarization as a sentence-selection problem. Sentences may be selected based on scores computed from the presence of topic-words or word-frequencies (Nenkova & Vanderwende, 2005). Centroid-based methods try to select sentences such that the resulting summary is close to the centroid of the input documents in the representation space (Radev et al., 2004). Rossiello et al. (2017) extend this approach by mapping sentences to their representation using word2vec embeddings rather than using TF-IDF weights. The main disadvantage of extractive methods is their limitation in copying text from the input, which is not how humans summarize. Banko & Vanderwende (2004) in particular found human-authored summaries of multiple documents to be much more abstractive.

Liu et al. (2015) present a framework for doing abstractive summarization in three stages. First text is parsed to an Abstract Meaning Representation (AMR) graph; then a graph-summarization procedure is carried out, which extracts an AMR sub-graph; finally, text is generated from this sub-graph. All three components require separate training and also AMR annotations, for which there is very little data. It is unclear how to generalize this to the multi-document setting.

Miao & Blunsom (2016) train an auto-encoder to do extractive sentence compression and combines it with a model trained on parallel data to do semi-supervised summarization.

Recently, there has been progress in learning to translate between languages using only unpaired example sentences from each language (Artetxe et al., 2017; Lample et al., 2017). Gomez et al. (2018) train CycleGan-like (Zhu et al., 2017) models to map between unpaired examples of ciphertext and decrypted-text. In contrast to this line of interesting work, we only have examples of the input sequence and thus cannot apply such techniques.

Review summarization systems have been designed with domain-specific choices. Liu et al. (2005); Ly et al. (2011) focus on producing a highly-structured summary consisting of facets and example positive and negative sentences for each. Zhuang et al. (2006) incorporate external databases to construct simiarly structured, movie-specific review summaries. In contrast our summaries have no explicit constraints or templates.

## 4 EXPERIMENTAL SETUP

### 4.1 DATASETS

We tuned our models primarily on a dataset of customer reviews provided in the Yelp Dataset Challenge, where each review is accompanied by a 5-star rating. We used a data-driven, wordpiece tokenizer (Wu et al., 2016) with a vocabulary size of 32,000 and filtered reviews to those with tokenized length, $L \leq 150$. Businesses were then filtered to those with at least 50 reviews, so that every business had enough reviews to be summarized. Finally, we removed businesses above the $90^{th}$ percentile in review count in order to prevent the dataset from being dominated by a small percent of hugely popular businesses. The final training, validation, and test splits consist of 10695, 1337, and 1337 businesses, and 1038184, 129856, and 129840 reviews, respectively.

### 4.2 EXPERIMENTAL DETAILS

The language model, encoders, and decoders were multiplicative LSTM's (Krause et al., 2016) with 512 hidden units, a 0.1 dropout rate, a word embedding size of 256, and layer normalization (Ba et al., 2016). We used Adam (Kingma & Ba, 2014) to train, a learning rate of 0.001 for the language

Table 1: Yelp results with $k = 8$ reviews being summarized. Note for Best/Worst Review WO scores: we exclude the best/worst review when calculating the average. Numbers are not provided for models that degenerated into non-natural language. As noted earlier, the NLL's are only provided for our abstractive models.

| | Model | Rating Accuracy | Word Overlap | NLL |
|---|---|---|---|---|
| | **Abstractive (ours)** | **52.09** | 26.48 | 1.19 |
| *Baselines* | No training | 24.44 | 19.68 | 1.29 |
| | Extractive (Rossiello et al., 2017) | 42.95 | 28.59 | – |
| | Best review | 38.48 | 23.86 | – |
| | Worst review | 30.01 | 13.14 | – |
| | Multi-Lead-1 | 40.69 | **31.64** | – |
| *Model Variants* | No pre-trained language model | 48.97 | 23.67 | **1.14** |
| | No auto-encoder | – | – | – |
| | Reconstruction cycle loss | 43.65 | 22.26 | **1.14** |
| | Early cosine loss | 19.32 | 14.28 | 1.71 |
| | Untied decoders | – | – | – |
| | Untied encoders | 50.89 | 26.29 | 1.20 |

model, a learning rate of 0.0001 for the classifier, and a learning rate of 0.0005 for the summarization model, with $\beta_1 = 0.9$ and $\beta_2 = 0.999$. The initial temperature for the Gumbel-softmax was set to 2.0.

One input item to the language model was $k = 8$ reviews from the same business or product concatenated together with end-of-review delimiters, with each update step operating on a subsequence of 256 subtokens. The initial states were set to zero and persisted across update steps for that set of $k = 8$ reviews in order to simulate full back-propagation through the entire sequence. The review-rating classifier was a multi-channel text convolutional neural network similar to Kim (2014) with 3,4,5 width filters, 128 feature maps per filter, and a 0.5 dropout rate. The classifier achieves 72% accuracy, which is similar to current state-of-the-art performance on the Yelp dataset.

# 5 RESULTS

## 5.1 MAIN RESULTS

The metrics for our model and the baselines are shown in Table 1. We find that the abstractive model outperforms all the baselines in rating accuracy. It also obtains a slightly lower, but comparable Word Overlap compared to the extractive method.

The human evaluation results are shown in Table 2 and validate our use of the automatic metrics in guiding our model development. The human scores on sentiment and information agreement are comparable for the extractive and abstractive models and rank order the methods similarly to our proxy metrics, rating accuracy and word overlap, respectively. We find that the abstractive summaries are comparable to extractive summaries and randomly selected input reviews on fluency, suggesting the model outputs have high fluency. We also find that the early cosine loss model has much lower ratings on the fluency questions, which agrees with the much higher NLL compared to our best-performing abstractive model.

An example set of reviews and corresponding summaries are shown in Figure 2. We find that extractive summaries, while highly specific and fluent, appear to summarize only a subset of the reviews. The abstractive summaries tend to be more general (e.g. using the term "mani/pedi" which does not occur in the input), but as the higher rating accuracy suggests, also more reflective of the average sentiment of the reviews. Because the model has no attention, there is very little copying and the summaries are highly abstractive. For summaries over the test set, 78.43% of 2-grams, 96.57% of 3-grams, and 99.33% of 4-grams in the summaries are unique (i.e. not found in the reviews being summarized). Figure 3 shows summaries of negative, neutral, and positive reviews from the same

Table 2: Mechanical Turk results comparing different methods.

| Model | Sentiment | Information |
|---|---|---|
| Abstractive (ours) | **3.91** | 3.83 |
| Extractive | 3.87 | **3.85** |

| Model | Grammar | Non-redundancy | Referential clarity | Focus | Structure and Coherence |
|---|---|---|---|---|---|
| Abstractive (ours) | **3.97** | 3.74 | **4.13** | 4.10 | **4.02** |
| Extractive | 3.86 | 3.93 | 4.05 | 4.01 | 3.99 |
| Early cosine loss | 2.02 | 1.84 | 2.02 | 1.96 | 1.95 |
| Random review | 3.94 | **4.06** | 4.09 | **4.23** | 4.01 |

---

**Original Reviews: Mean Rating = 4**

No question the **best pedicure** in Las Vegas. I go around the world to places like Thailand and Vietnam to get beauty services and this place is the real thing. Ben, Nancy and Jackie took the time to do it right and **you don't feel rushed**. My cracked heels have never been softer thanks to Nancy and they didn't hurt the next day. </DOC> Came to Vegas to visit sister both wanted full sets got to the salon like around 4 . **Friendly** guy greet us and ask what we wanted for today but girl doing nails was very rude and immediately refuse service saying she didn't have any time to do 2 full sets when it clearly said open until 7pm! </DOC> This is the most clean nail studio I have been so far. The service is great. **They take their time** and **do the irk with love**. That creates a very **comfortable atmosphere**. I recommend it to everyone!! </DOC> Took a taxi here from hotel bc of reviews -Walked in and walked out - not sure how they got these reviews. Strong smell and broken floor - below standards for a beauty care facility. </DOC> The **best** place for pedi in Vegas for sure. My husband and me moved here a few months ago and we have tried a few places, but this is the only place that makes us 100% happy with the result. I highly recommend it! </DOC> This was the **best** nail experience that I had in awhile. The service was perfect from start to finish! I came to Vegas and needed my nails, feet, eyebrows and lashes done before going out. In order to get me out quickly, my feet and hands where done at the same time. Everything about this place was excellent! I will certainly keep them in mind on my next trip. </DOC> I came here for a munch needed **pedicure** for me and my husband. We got **great customer service** and an amazing **pedicure and manicure**. I will be back every time I come to Vegas. My nails are beautiful, my skin is very soft and smooth, and most important I felt great after leaving!!! </DOC> My friend brought me here to get my very first **manicure** for my birthday. Ben and Nancy were so **friendly** and super attentive. Even though were were there past closing time, **I never felt like we were being rushed or that they were trying to get us out the door**. I got the #428 Rosewood gel **manicure** and I love it. I'll definitely be back and next time I'll try a **pedicure**.

**Extractive Summary: Predicted Rating = 1**

Came to Vegas to visit sister both wanted full sets got to the salon like around 4 . Friendly guy greet us and ask what we wanted for today but girl doing nails was very rude and immediately refuse service saying she didn't have any time to do 2 full sets when it clearly said open until 7pm!

**Unsupervised Abstractive Summary: Predicted Rating = 5**

Probably the **best mani/pedi** I have ever had. I went on a Saturday afternoon and it was busy and they have a great selection of colors. We went to the salon for a few hours of work, but this place was **very relaxing**. **Very friendly staff** and a great place to relax after a long day of work.

Figure 2: An example of input Yelp Reviews (separated by "</DOC>") with the extractive baseline and our model summaries. Certain words are colored in the original reviews that correspond to similar words in the abstractive summary.

business, allowing us to see how the summary changes with the input sentiment. More examples can be found in Appendix D.

---

**Summary of Negative Reviews: Predicted Rating = 1**
Never going back. Went there for a late lunch and the place was packed with people. I had to ask for a refund, a manager was rude to me and said they didn't have any. It's not the cheapest place in town but it's not worth it for me. And they do not accept debit cards no matter how busy it is. But whatever, they deserve the money .

**Summary of Neutral Reviews: Predicted Rating = 3**
Food is good and the staff was friendly. I had the pulled pork tacos, which was a nice surprise. The food is not bad but certainly not great. Service was good and friendly. I would have given it a 3 star but I'm not a fan of their food. Service was friendly and attentive. Only complaint is that the staff has no idea what he's talking about, but it's a little more expensive than other taco shops.

**Summary of Positive Reviews: Predicted Rating = 5**
Always great food. The best part is that it's on the light rail station, and it's a little more expensive than most places. I had a brisket taco with a side of fries and a side of corn. Great place to take a date or to go with some friends

---

Figure 3: Summaries generated from our model for one business, but varying the input reviews. Each summary is for a set of reviews with the same rating. The original reviews are found in the appendix (Figure 12).

## 5.2 MODEL VARIANT ABLATION STUDIES

The results of the ablation studies are shown in Table 1.

The language model experiments indicate that while initializing the weights with a pre-trained language model helps, as has been shown in sequence-to-sequence models (Ramachandran et al., 2016), it is not critical. Without the pre-trained language model, the rating accuracy and average WO score are only a few points lower (48.97 vs. 52.09 accuracy, 23.68 vs. 26.48 WO). We also find that using a pre-trained language model alone, without training the summarization model, is not enough to generate good summaries. While the generated texts are fluent, they fail to actually summarize the reviews, as shown by the low rating accuracy (24.44) and WO score (19.68).

Two of the models failed completely, with the model degenerating from producing natural language (even though initialized with the pre-trained language model) to garbage text. The first variant, without the auto-encoder, converges to a trivial solution – the cosine similarity loss can be minimized if the encoders learn to produce the same representation $z_j$ regardless of the input. As suspected with the second variant, in which the decoders are not tied, the summary decoder has no constraint to remain in language space, $\mathbb{V}$.

The reconstruction cycle loss works, but worse than the original model (43.66 vs. 52.09 accuracy, 22.26 vs. 26.48 WO). We hypothesize its lower performance is due to difficulties in optimization. Although the Gumbel softmax trick allows the model to be fully differentiable, the gradients will have either high bias or or variance depending on the temperature (which can be annealed during training). With the single loss function being after the Gumbel softmax step, the model may be difficult to optimize. We also believe that reconstructing the original texts from $\phi_E(s)$ is difficult, as $s$ is a lossy compression of the original documents.

Next, we see that the "Early cosine loss" model has poor rating accuracy and average Word Overlap. We believe this is largely due to exposure bias, resulting in a relatively large NLL. The summaries are generated at test time through greedy decoding, but this decoding process (and thus the summary decoder) is not part of the training procedure. Critically, the full model does not suffer from this problem. Manual inspection of the samples confirm the summaries are disfluent.

Finally, the model performed approximately as well without tying the encoders (50.89 vs. 52.09 accuracy, 26.29 vs. 26.48 WO). Given that this is the case, it makes sense to simply tie the encoders and reduce the number of model parameters.

We also examined the fluency of each model by plotting the negative log-likelihood of the generated summaries during training, as shown in Figure 4. The two models that fail are immediately evident, as indicated by their large NLL's.

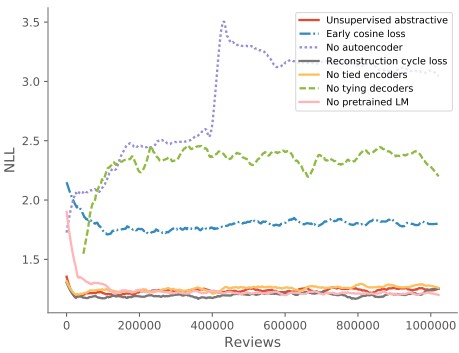

Figure 4: Negative Log-Likelihood of Models During Training

## 5.3 VARYING $k$

We also ran experiments to examine the effect of different $k$'s and whether our model was robust to this hyperparameter. The results for training and testing with different $k$'s are shown in Figure 5 and Figure 9. The "Varying $k$" model was trained with $k \in \{4, 8, 16\}$ reviews (randomly selected every minibatch). We did not perform any extra hyperparameter search for the models trained with $k = 4$, $k = 16$, and varying $k$, and instead used the same hyperparmeters used to train the $k = 8$ model.

Overall, we find that the rating accuracies are largely stable for all methods. Similarly, we find that the abstractive model also has relatively stable Word Overlap scores, with a small decline as the number of reviews being summarized at test time increases. However, the extractive baseline method appears to decrease as $k$ increases – as the variance in content increases with greater $k$, it may be harder to extract sentences that reflect all of the reviews.

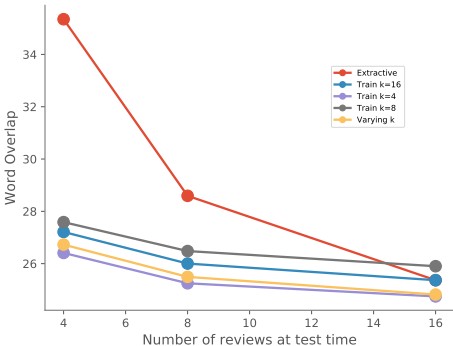

Figure 5: Word overlap score with varying $k$ at train and test time

## 5.4 AMAZON DATASET

To check that our model works beyond Yelp reviews, we also tested on a Amazon dataset of product reviews. We selected two different categories – Movies & TV and Electronics. We used the same parameters used to filter the Yelp dataset, resulting in training, validation, and test splits of 6,237, 780, and 780 products, and 583776, 73040, 73040 reviews, respectively. No tuning was performed – we used the exact same model and training hyperparameters as the Yelp models. The results are shown in Table 3. We find similar results – the abstractive method outperforms the baselines in rating accuracy and has slightly lower Word Overlap than the extractive baseline method.

Table 3: Results on Amazon dataset

| Model | Rating Accuracy | Word Overlap | NLL |
|---|---|---|---|
| **Abstractive (ours)** | **47.90** | 27.02 | **1.23** |
| No Training | 38.04 | 18.10 | 1.37 |
| Extractive (Rossiello et al., 2017) | 43.86 | 30.41 | 1.38 |
| Best review | 45.05 | 24.59 | 1.29 |
| Worst review | 38.88 | 13.79 | 1.36 |
| Multi-Lead-1 | 44.90 | **32.18** | 1.33 |

## 5.5 QUALITATIVE ERROR ANALYSIS

Although most summaries look reasonable, there are occasionally failure modes. We discuss the common failure modes as follows, with examples in Appendix D.

**Fluency errors:** Grammatical mistakes (e.g. incorrect use of 'and' or 'but') and repetition of phrases could perhaps be reduced with a more powerful language model, as state of the art language models use many more parameters.

**Factual inaccuracy**: Summaries sometimes make reference to named entities (e.g. the city the restaurant is located in) that are incorrect or not found in the original reviews. Factual accuracy is an ongoing area of research in the summarization field, and a loss function to penalize inaccurate statements may be helpful here.

**Rare categories:** Summaries appear to be worse for categories with limited data (e.g. parks in the Yelp dataset). This could potentially be addressed by up-sampling these reviews or fine-tuning towards specific categories.

**Contradictory statements** This sometimes occurs when there are both highly positive and highly negative reviews, leading to a summary with a positive statement immediately followed by a negative statement about the same subject. We could either separate these statements in a post-processing step, or train a model that is conditioned on the rating (and possibly other latent variables).

## 6 CONCLUSION, LIMITATIONS, AND FUTURE WORK

The standard approaches to neural abstractive summarization use supervised learning with many document-summary pairs that are expensive to obtain at scale. To address this limitation and make progress toward more widely useful models, we introduced an unsupervised abstractive model for multi-document summarization that applied to reviews is competitive with existing unsupervised extractive methods. [3]

The proposed model is highly abstractive because it lacks attention or pointers – future work could incorporate these mechanisms to provide summaries that contain both the most relevant points and the specific details to support them.

For our problem, summarizing multiple documents in the form of a similarly distributed single-document was appropriate, but may not be in all multi-document summarization cases. Learning to tailor the summary to a different desired form (with few examples) would be an interesting extension.

Our model does not provide an unsupervised solution for the more difficult (as there are fewer redundancy cues) single-document summarization problem. Here too there are extractive solutions, but extending ideas presented in this paper might yield the similar advantages of neural abstractive summarization.

---

[3]Code to reproduce results will be available soon on Github.

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

## APPENDIX A    MODEL VARIATIONS

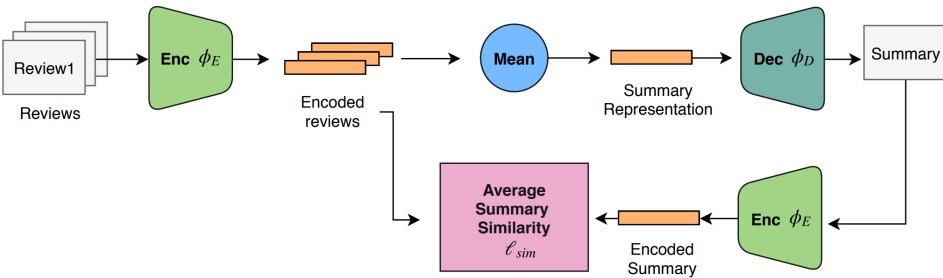

Figure 6: Model Variant: No Autoencoder

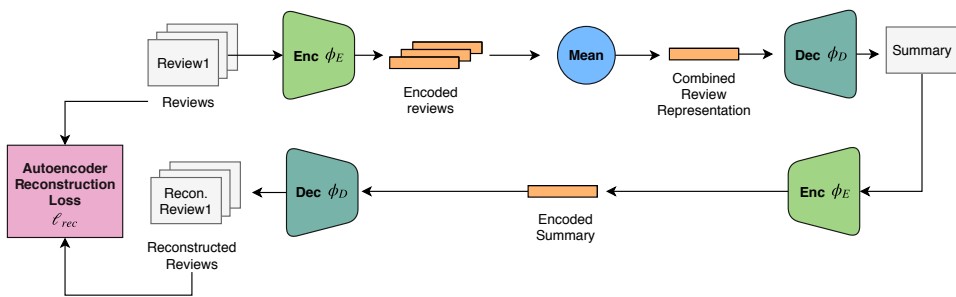

Figure 7: Model Variant: Reconstruction Cycle Loss

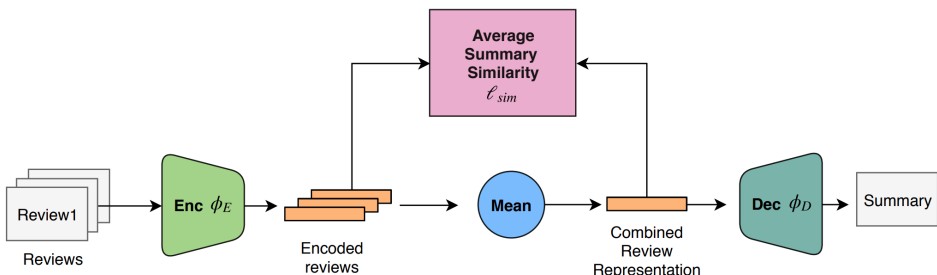

Figure 8: Model Variant: Early Cosine Loss

## APPENDIX B    EXPERIMENTAL SETUP

### B.1    HUMAN EVALUATION

The results were collected through two separate experiments on Mechanical Turk. For both experiments, the order of the models were randomized so as to prevent ordering biases (e.g. summaries from the first model receiving higher ratings). To obtain higher-quality responses, workers were selected with the following criteria: (1) Masters Qualification (granted by Mechanical Turk depending on a worker's performance across tasks over time), (2) at least a 95% HIT approval rate, (3) at least 100 HIT's approved, and (4) worker's location is the United States.

In the first experiment, in which workers were asked to evaluate how well the summary reflected the information and sentiment of the original reviews, we presented the eight reviews being summarized and the extractive and abstractive summaries next to them.

In the second experiment, in which workers were asked to evaluate the fluency of the summaries, we presented summaries from four different models. The original reviews were not shown, as the questions were simply assessing the linguistic quality of the summaries themselves.

The definitions of the five dimensions of fluency were taken from Dang (2005) and defined as follows:

1. Grammaticality: The summary should have no datelines, system-internal formatting, capitalization errors or obviously ungrammatical sentences (e.g., fragments, missing components) that make the text difficult to read.

2. Non-redundancy: There should be no unnecessary repetition in the summary. Unnecessary repetition might take the form of whole sentences that are repeated, or repeated facts, or the repeated use of a noun or noun phrase (e.g., "Bill Clinton") when a pronoun ("he") would suffice.

3. Referential clarity: It should be easy to identify who or what the pronouns and noun phrases in the summary are referring to. If a person or other entity is mentioned, it should be clear what their role in the story is. So, a reference would be unclear if an entity is referenced but its identity or relation to the story remains unclear.

4. Focus: the summary should have a focus; sentences should only contain information that is related to the rest of the summary.

5. Structure and Coherence: The summary should be well-structured and well-organized. The summary should not just be a heap of related information, but should build from sentence to sentence to a coherent body of information about a topic.

## APPENDIX C    VARYING $k$'S EFFECT ON CLASSIFICATION ACCURACY

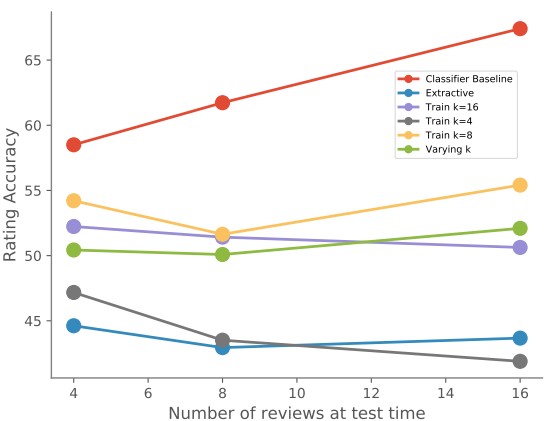

Figure 9: Classification accuracy with varying $k$ at train and test time

## APPENDIX D    EXAMPLES

### D.1    ADDITIONAL EXAMPLES

**Original Reviews: Mean Rating = 4**
**Crepe on point!** Just the way it is supposed to be. **Owner very patient and has great customer service**. Will return. </DOC> **Small place** but **friendly** cheap tasty and fast! Sat on the barstool by the window with my gf and we were enjoying our time there. Would recommend to anyone looking for a decent breakfast </DOC> The **crepes were pretty good but** my 4 and 7 year old preferred the ones that I make at home. **Service was good** though and coffee was above average. Breakfast for 4 came out to $47 which was a bit much considering I spent only a fiver more for a killer dinner for four last night at Amelio's. </DOC> We had a **ham and cheese crepe as well as a Nutella, strawberry, and banana one, and they were great**. The **crepe was tasty** and chewy, as well as the fillings. **Each crepe** also came with a since sized portion of fresh fruit. Worth a try if you are looking for a **crepe** near downtown. </DOC> The **service is friendly**. It's a **small, homely place**. **Great crepes** at a reasonable price. </DOC> **Quaint little shop** halfway under ground, colorful inside. Only two workers when we went. Lots of options, both sweet and savory. My husband got the thyme and sesame seed **crepe, which was amazing**. I got the spinach and egg crepe. It was quite boring, but the original had cheese on it and I had them hold the cheese, so that's my fault. I'd definitely like to come back for a sweet crepe or perhaps a thyme crepe all for myself :-) </DOC> favorite one, surprised me every time. Starter and choco-strawberry are great choices. Surprise is great too! </DOC> I ordered the **crepes with nutella and strawberries and it was honestly so good**. And I love the place as well, it's **small yet cozy**. My new go to breakfast place because it's so close to my house and pretty cheap for such **delicious crepes.**

**Extractive Summary: Predicted Rating = 5**
I'd definitely like to come back for a sweet crepe or perhaps a thyme crepe all for myself :-) favorite one, surprised me every time. My new go to breakfast place because it's so close to my house and pretty cheap for such delicious crepes.

**Unsupervised Abstractive Summary: Predicted Rating = 4**
The **crepes and** service **are great**. My only complaint was that the **seating was limited, so it could be a little more intimate**. **Service was friendly and attentive**. I'll be back to try out other items on their menu.

Figure 10: Extractive and abstractive summaries are both classified positive.

## D.2 MODEL COMPARISON

**Original Reviews: Mean Rating = 4**
This spot is amazing i dont understand negative reviews such a great spot with authentic food loved it. </DOC> This place is dirty. There are roaches in the dining and restroom areas. They refused to refund my order. The people who work there are rude and useless. There is no reason to waste your time or money on eating here. </DOC> I come to this place a lot. Sam is the man and the quality of lamb kabob is amazing. You can't beat the price. Plus they got free Wi-Fi and plush couches to lounge on afterwards and have some tea and do some work. I highly recommend it. </DOC> Talk about friendly customer service! From the moment we walked in, we were greeted nicely and immediately felt welcome. I came here with my baby girl and boyfriend. We tried the beef, chicken and lamb kabobs. All were great but my favorite by far was the ground beef. It was so tender, flavorful and delicious! The freshly made bread and hummus was great too. Even my baby girl loved it. We will definitely be back again for more! </DOC> First time i stopped by i tried the chicken shawrma also the appetizer sampler ( 3 flafel, humus and baba ghanosh ) and the bread was made fresh worth to give it a shot </DOC> This place is amazing and their food is out of this world!! The food is so good and fresh! Customer service is great since its under a new management !!! Love the people that work there! Cant wait to go with my friends there for their hookah nights!! </DOC> I've had gyros in many places but I can sincerely tell you that this place makes the best gyros I have ever tasted and The Baklava delicious I will definitely return to this place and recommend it to anyone. </DOC> Lunch: Beef & lamb shawarma. Comes with pita, hummus, tzatziki, salad and...onion salad? with lemon wedge. Meat was a bit on the tough side & heavily seasoned. A bit spicy–guess I'm a wimp today. Needed the tzatziki. Other than that I enjoyed it.

**Extractive Summary: Predicted Rating = 5**
First time i stopped by i tried the chicken shawrma also the appetizer sampler ( 3 flafel, humus and baba ghanosh ) and the bread was made fresh worth to give it a shot This place is amazing and their food is out of this world!!

**No Training Summary: Predicted Rating = 4**
I've only come here for lunch. The soup is always fresh and seasoned deliciously, always packed full of flavor. Service is spotty and sometimes they rush you out of the line at the bar. Open kitchen is clean and neat, and the servers are always nice.

**Early Cosine Loss Summary: Predicted Rating = 1**
This place is so good and the food. And my only the places, Chicago, i go the meat the Chicago places always always deli deli always always deli deli always the grocery, the grocery is always clean and the Best always always clean and the lamb with the lamb (i always with lamb meat with the lamb meat i the lamb (with the lamb lamb (i the the the ( (the always the (the the the grocery (the the the Best (the always the Best (i always the the most the the grocery, the the the Best grocery i the the Best the the the the the today today today today today today today today today today today today today today today today today today today today today today today

**Unsupervised Abstractive Summary: Predicted Rating = 5**
Some of the best Mediterranean food I've had in a long time. My favorite is the chicken shawarma and it was so tasty!! Also their homemade pita bread is a must try! If you are looking for a place to take your business, make sure you check out this place as it is a lot of food.

Figure 11: Example summaries from different baselines and model variations

## D.3   ONE BUSINESS, DIFFERENT REVIEWS GROUPED BY RATING

I usually don't write reviews unless It is terrible. One word horrible , longest wait ever and considering how Hungry u would Think the food tasted better. when we received our chicken pork and brisket tacos it was sooo dry. I don't understand How they call this place a bbq Because it tastes nothing like it. I will never ever come Here again. </DOC> Impossible to find. Snooty hipster waitstaff. Cash only and a $4 fee arm. I don't care how good your food is. </DOC> Prices have gone from $4.75 & up for one taco to $6 & up. Not worth it. Go to Las Palmas or Doce instead. </DOC> Made a reservation, was seated 30 minutes late. Waited 15 minutes for a server. Got the rib plate, dry on the inside and greasy on the outside. Also was scolded for looking at the gluten free menu for some reason. But that's ok not going back. Thanks I see why you are one of the lower reviewed establishments in the city. I'll do my part as well in contributing to that. </DOC> hipster hell!!!! horrible obscure music, skinny jeans everywhere, crappy food, and no heat!!!! servers had disgusting nose piercings. only go to this place if you love participation trophies and bernie sanders </DOC> Why would you have at the top of your website "NO RESERVATIONS" and then when we show up ask if we called ahead and then tell us it will literally be a three hour wait to get in? Oh I didn't call to get on the list? Why the hell would I think there was a list your site straight up said no reservations so why would I bother to call? Lawrenceville sucks now anyhow. </DOC> Ordered a captain and Coke. Was informed they don't have captian and.They don't have Coke. Told them to make as best they cold. Got the worst watered down rum and Coke I've ever had and got charged $30 for the 3 pathetic drinks we ordered. Absolute worst and I've lived in several areas even NYC. </DOC> My first experience was good. The food was above average, but the wait time was pretty long. Went for a 2nd visit for lunch today and ordered two tacos but had to leave before eating, because the order still hadn't come after 35 minutes! The waitress wasn't very nice when asked about the delay in serving my order. The place was only half full. Maybe others have had the same experience I had and made the same decision not to go back

(a) Negative Reviews: Rating = 1

Great for take out but atmosphere is a low point. We hadn't been to Smoke since it was in Homestead so I was ready for a yummy taco. The food did not disappoint. Both my favorites, brisket and chicken apple were delicious. Service was fine. But the music was obnoxiously loud combined with ambient noise to the point that conversation was impossible and digestion questionable. As I walked past the kitchen I noticed it was quieter there. We took our order to go. The new place looks hip but I couldn't enjoy it. Sorry we couldn't stay because we really like you guys. Missing the quiet little place in Homestead that was all about good food. </DOC> I think I could've read a Russian novel in the time between when I placed my order and received my food. Seriously, that was a crazy long wait to get the food. We're talking about a few tacos, it really shouldn't take that long. And, our seating area was drafty. Plus the music was too loud. And for what you get, it's a bit pricey. However, the tacos are quite tasty. Three of the four of them were very good (the chorizo was so-so). The pork was indeed smoky; the chicken was appealing too. If they could improve their operation in the other areas, they'd be a four star place. </DOC> This was our first visit to Smoke during a weekend trip. Cool, unique neighborhood. The inside of the place is a simple and rustic but welcoming. Service was friendly. It is cash only which I find to be a nuisance... Also BYOB, at least for now. We started with the bowl of cheese. It was delicious but we agreed that half the portion at half the price would be more suitable for two people. The tacos were good. We had chicken, pork, and brisket. I enjoyed the bbq flavors but found the tacos to be each a little one-noted for 6-7 bucks a pop. </DOC> I went on burger night therefore the menu was limited but it was nice and they had a few non-burger options. Also there was no wait for a table around 8pm. I had the appetizer fries with brisket and it was more than enough food to fill me up. I also had the Big Fiz tobdrink (St. Germaine and grapefruit cocktail) and found it to be so light and refreshing. It's a great summer cocktail and perfect to lighten up a big heavy meaty meal. </DOC> A couple of years ago this place would have been awarded a five star, but I'm afraid it's gone down hill. They reduced their taco options and expanded into burgers and plated meals. We decided to sit at the bar and bypass a 30+ minute wait. I ordered the brisket taco and pork taco. The meat in the tacos was dry and overwhelming with sauce, it poured out on the tray while eating them. The mac and cheese side was bland and lacked salt. Maybe we just hit this place on a bad night. I will go back again to make sure, but this trip was disappointing. </DOC> Decent food. Great service. Menu is hit or miss depending on the day you go. Great idea but inconsistent food. </DOC> Overhyped for the price. Solid food and half decent service. I tried it out on a whim and it wasn't all that it was hyped to be. The best part of the place is the smell of the food cooking. If you enjoy the hipster beard crowd this is the place for you. Not bad but nothing that makes me want to go out of my way to revisit. </DOC> Great tacos and queso, atmosphere is cute, but the wait is intolerable and the staff is unfriendly, which took away from the experience.

(b) Neutral Reviews: Rating = 3

My second visit and just as impressed! The service has been awesome and they have been more than willing to accomodate me and my food allergies/restrictions even on a busy Friday night. So excited to be living right around the corner! </DOC> Best Tacos in Pittsburgh seems like wan praise. Like...prettiest girl in the trailer park. And I don't have a TON of experience with taco places (or trailer park girls), but I have some...and this is the best one. I try to avoid referencing other places relative to a place I'm rating, but suffice it to say that Smoke has competition, but as far as a more or less conventional ingredient taco goes...Smoke is it! (like Coke is it...see what I did there?) Nice beer selection. Very cool decor. Great location. Friendly servers. Great food. Win. </DOC> This food is so delicious. The best thing on the menu is definitely the queso. You absolutely cannot skip it. Prices are pretty reasonable. Only negative is that you always have to wait a super long time to get a table. </DOC> Great lunch today at Smoke. Good craft beer list on tap to start things out. The special today was a smoked mushroom taco which was exceptional. The brisket was tasty but the mac and cheese was to die for. </DOC> I dream about their chips and queso! Great tacos and plenty of options for vegetarians. I only wish there was a location closer to my house so I could go more often. </DOC> We have always enjoyed the food and vibe at this taco spot. We used to go to their location in Homestead often but are happy they are now located in Lawrenceville. BYOB is a plus, but don't forget it's cash only. I would recommend any of the tacos, they are all great. Also, we ordered the queso and chips. Absolutely amazing. Soft fried pita chips are so good. Eat it! </DOC> Awesome food, run don't walk. Generous portions, only downside was that dessert was crazy expensive. Maybe they could let you known in advance that the "Pie" is small and perfect for 2 people to share, but they will be charging you $12. </DOC> Love this place. Great offerings with a barbecue twist. Best Mac n cheese I have ever tasted. Great service. Nothing bad to say.

(c) Positive Reviews: Rating = 5

Figure 12: Reviews grouped by rating

## D.4 QUALITATIVE ERROR ANALYSIS

---

**Original Reviews: Mean Rating = 4**

First visit to this great little diner today......the crab Benedict was laden with huge chunks of tasty crabmeat and accompanied by great hashed potatoes. Every item ordered at the table was perfect! The server was prompt and helpful and the place has that roadside diner ambiance although it's in the heart of the city. I'll be back in a few days, for sure. </DOC> This place was great. Great food, great service, and great atmosphere. Sat at the bar and got to watch the staff in action. True team effort. Everyone was happy to be there and happy to help...and it showed in the food. Will definitely be back and definitely recommend. </DOC> No knock against the food, it was very straight forward. The shredded hash browns were very bland, will need some sort of seasoning to eat them. Serves was very friendly as soon as we walked in. Was not able to accommodate my egg allergy when I asked if I could supplement something for the eggs that came with my country fried steak. Not getting anything higher unless they can go above and beyond. Fine but nothing remarkable. Also way too expensive, with ripe 35 for 2 peoples... Cmon it's a dinner... Stop it. </DOC> Kelly's actually catered a wedding I was at tonight and the food was fantastic! There was carrot ginger soup, mushroom risotto balls and create your own pasta. I had linguini, sausage, spinach, mushrooms and red onion with alfredo sauce. To Die For!!! Seriously one of the best things I have ever eaten. Really want to go to the diner to check out the full menu. I give the catering an A ++++ </DOC> This breakfast spot will make you want to come back day after day. Your heart and gut may not agree - but who cares! The breakfast bagel with hot sausge patty is my go to. I love the price, the atmosphere and the food even more. It's not fancy, but it really doesn't have to be. </DOC> Been here 2x for breakfast and 2x for lunch in last 2-3 months. Breakfast...had variety of different things. I would stick with anything eggs and meat. Had pancakes once and were a little g̈reasyf̈or me. Portions are BIG. If you leave full, shame on you. Lunch... fish sandwich - good tuna melt - real good (friend had it - tasted it - yummy) burger club - good (tried zucchini fries with it which were real good and portion was HUGE) For what it is...diner, it is well worth the stop!!! </DOC> Every time, I mean every time, Kelly O's hits the spot. I must say I am an early bird and hit the breakfast joints before 8 on the weekend. I wasn't really, really hungry, but they have a lil one breakfast that is one egg, one meat, some potatoes and Mancini toast. Just enough. I added a pancake for a taste of sweet. For the heartier eater, there are daily specials and a variety of large bfast plates. The key to Kelly O's is that the service is always prompt and icing on the cake is mancini toast. </DOC> Kelly-Os is a great Pittsburgh diner. The food is fairly priced, delicious, and fast. Their fresh-squeezed orange juice is the best orange juice ever. The French toast isn't anything special (and not recommended), but the pancakes are nice and fluffy. This is one of my boyfriend's favorite places to get Eggs Benedict **in Pittsburgh**. Some of the waiters/waitresses are nicer than others, so one thing they can work on is better overall friendliness. However, I always have a nice time with good company and great food.

**Unsupervised Abstractive Summary: Predicted Rating = 4**

Last time I was here, the server was really nice and helpful. For breakfast it's a **great place to eat, eat, breakfast or lunch**. The place is a bit small but it's not too far from home. Dinner for two of us was amazing. **I had a side of mashed potatoes and gravy with a side of potatoes and gravy**. Both were delicious and the onion rings were also very good. A great spot to eat in **downtown Phoenix.**

---

Figure 13: Example of repetitiveness and factual inaccuracy (restaurant is in Pittsburgh)

**Original Reviews: Mean Rating = 4**
Affordable, efficient and always do a great job. Even my boyfriend got his brows done here once. Highly recommended! </DOC> Needed legs and lady parts taken care of in a jiffy. this place was near home and priced reasonably. I was looking for somehwere new to replace the closer establishments on bloor where front of house welcome is underwhelming and treatments rushed and often not that great. Naheed was awesome. She was attentive and thorough and was a real sweetie. when I mentioned I'd never had an eyebrow threading before, she began to explain the process and before I knew it, she gave me my first threading at no extra charge! So happy. Personable experience that had me walking away feeling good. Thank you Naheed! </DOC> My eyebrows got butchered from a threading shop on Gerrard (in little India) so I worked extremely hard to grow them out. After reading some of the other reviews, I decided to check this place out and I was NOT disappointed. Hamilda was the one that reshaped and cleaning up my brows. She did a great job and really knew what she was doing. This will be my new spot for eyebrows! </DOC> After reading various positive reviews here on Yelp, I scheduled a morning appointment. Naheed, the owner, welcomed me as her first customer of the day. It was my first time getting eyebrow threading done. She was quick, professional and approachable, and made me feel welcome and comfortable. We actually also did a wax as well. Which was thorough. It's best to get an appointment, as there were walk-ins waiting after I was finished. Thanks again Naheed, I plan to come back again when I'm due for one of your services again, you deserve 5 stars. </DOC> Love this place! Both Naheed and Nadia are amazing at what they do! The place is clean and walk-ins are always welcome. The prices are super reasonable and there's always deals on for even more saving! So far I've only been for waxing and haven't left with any complaints. Gonna try out eyebrow threading next. </DOC> They did such a great job threading my eyebrows!! She charged about $11 for eyebrow threading. I forget the name of the lady that did my eyebrows :/ but she did such an excellent job! </DOC> The most amazing threading place ever! Ive been trying to find a threading place in canada for so long and finally found my regular to go place! They do an amazing job shaping the eyebrow, they are really the experts!!! Defiantly torontos best threading place! </DOC> I don't recommend this place for eyebrows threading! She literally butchered my eyebrows and wasn't receptive at all. She works way too fast and is not meticulous at all. Too bad!

**Unsupervised Abstractive Summary: Predicted Rating = 5**
My brows were beautiful and well done. **It's the closest place to my house for me, but I'm not sure if it is a sign or a very small place. But when it was time to sign up for the appointment, it's great people!** and they are great for brows and feet!

Figure 14: Example of fluency problems

**Original Reviews: Mean Rating = 2**
Horrible service and please check your cat when you pick up from service, they have changed the wheels once and other time that have stolen the tools that a new car comes with. I will never buy a car from them anymore especially because the service department. </DOC> Was passing thru Las Vegas when my 2007 X5 4.8i had a belt tensioner failure. Limped to this dealer the next morning from my hotel and was kindly greeted by Larry. Years later...Horror strikes from the slipshod work this dealer did to my E70. </DOC> As many people here have mentioned this is not a place where you want to buy a car. DO NOT let the BMW name fool you, this is a sleezy used car dealership in disguise. The salesman are quick to sale you a car but will NOT get back to you when they deliver a car that was not promised. My car was missing a floor mat, the oil was not changed and has other issues. DON'T be fooled by the rep responding to bad reviews, thats just to save face. I have reached out several times and have yet to get a response. Save yourself the time and energy and buy a car from a professional dealership. DO NOT BUY from BMW of Las Vegas. </DOC> Bought a preowned BMW and their detail work is terrible. They left dirt in between buttons, and promised me that they would neutralize the perfume smell coming from the arm rest. I had to go several times to get this done, and it still wasn't done properly. On top of that there was a scratch they promised to repair and it never happened, they kept giving me odd times to go in during the week without a rental. They were not very understanding that I had to work and can't take days off for something that could of easily be fixed on their end. Never buy from them again, for them it's all about the $$$. </DOC> Took my car in for routine servicing and when I picked it up later that day I noticed that my front splash guards were missing after the service team took my car through the car wash. I informed the service manager and he assured me that my car came in without splash guards until, I pointed out that that they had thrown them in the front seat and they were still dripping wet from the car wash. He reluctantly admitted his team had torn them off in the car wash and put them in the front seat. Overall just a extremely shady operation. </DOC> awesome awesome awesome these guys are great i have never had such a great experience thanks Trent!!!! </DOC> Great car buying experience. Got a really good deal on my new BMW and was treated like royalty by all the professionals at Las Vegas BMW. I highly recommend them. </DOC> Thanks for hooking us up with a wonderful certified pre-owned Beamer.. Great dealership, BMW should be proud you all are representing them in Las Vegas.

**Unsupervised Abstractive Summary: Predicted Rating = 5**
**Took my car in for a Brazilian wax and transmission fluid.** They were very accommodating, the customer service was good and he was able to get me in the same day I called. He was very friendly and helpful in explaining what he was doing and what was going on in the morning. I will be going back to this dealership for sure.

Figure 15: Example of worse quality on rare categories (Categories: Automotive, Auto Parts & Supplies, Auto Repair, Car Dealers)

**Original Reviews: Mean Rating = 5**

My first time at a Dragon Pearl buffet. They had a huge selection of food from over the world and all tasted very good! My favourite was the deep fried oysters (make sure you choose ones with just a little batter). Unfortunately I didn't get to try their signature dragon pearl dessert...Friends told me that the Markham location had cold crab and frog legs, so I was looking forward to it at this location... they didn't have it! </DOC> Just one sentence : I love this restaurant over any restaurant not only in Toronto but over the country . Highly recommended because of many reasons, the food is such fresh that you can't find in even expensive restaurants . The atmosphere is very sexy and warm that you like to stay there for a long time chatting with you family and friends.on Tuesdays, there is a special promotion that is 11.99 $ for lunch and 1.5 $ for green tea per each person. I recommend the lobster in weekends which is given to you by a voucher . In sum, this restaurant put the Mandarin in real shame. Highly highly recommended. </DOC> This place is one of the best buffets that I have tried. They have a good variety of food and the service is amazing. On our second visit they were constantly clearing away places and replenishing napkins without asking. The roast beef was great and so was the fresh noodles, I highly recommend the beef szhewan </DOC> Great Chinese buffet in North Toronto. Prices are steep but a good pick for a lunch buffet </DOC> went there with friends for lunch. Food is average ,not impressive. renovation is really stylish though. </DOC> my parents really like this buffet. its actually pretty decent. the decor and unique and equisite and adds to the atmosphere. theres a strangeness to it it makes you feel like your on vacation or something. hard to explain. the food and variety is good. and lobster, well how can you complain. all.in all hapoy experience everytime ive been there. my only complaint would be theres this stupid rule about ordering tea and not being able to get more than one glass to share with the table. like what the fcuk is that? to me thats just cheap and how much tea can one family drink anyway, isnt the idea to fill up on food, why would they worry about people taking advantage of tea. lol. </DOC> Wow is the first word that comes to mind about this place. Talk about everthing you can get in a Chinese restaurant but with a twist, there is also a Sushi bar which loved sushi is one of my fav go to fast food. By far one of the BEST buffets i have been to in a while . I am going to sit here and enjoy this moment. While i dig my fork in a bowl of sticky rice and savor this moment. </DOC> The ambiance is great here, food is replenished more frequently than other buffet places, so you don't find food sitting out for two long and end up being dry and disgusting. Though I believe the sister restaurant offers a bit more variety.

**Unsupervised Abstractive Summary: Predicted Rating = 4**

Sushi is good and not too pricey. **I'm not a big fan of the food, but the food is great .** They have a nice selection of dishes and different presentations. I would recommend this place to anyone who likes spicy food but not in a hurry to get a good meal. Very good value for the money.

Figure 16: Example of contradictory statements

