# OpenReview forum: "Unsupervised Neural Multi-Document Abstractive Summarization of Reviews"
_ICLR.cc/2019/Conference_

### Official Review · AnonReviewer3 · 2018-10-31
**Novel work breaking ground on abstractive unsupervised multi-document summarization**

**Rating:** 9
**Confidence:** 4

**Review:**

# Positive aspects of this submission

- This submission presents a really novel, creative, and useful way to achieve unsupervised abstractive multi-document summarization, which is quite an impressive feat.

- The alternative metrics in the absence of ground-truth summaries seem really useful and can be reused for other summarization problems where ground-truth summaries are missing. In particular, the prediction of review/summary score as a summarization metric is very well thought of.

- The model variations and experiments clearly demonstrate the usefulness of every aspect of the proposed model.

# Criticism

- The proposed model assumes that the output summary is similar in writing style and length to each of the inputs, which is not the case for most summarization tasks. This makes the proposed model hard to compare to the majority of previous works in supervised multi-document summarization like the ones evaluated on the DUC 2004 dataset.

- The lack of applicability to existing supervised summarization use cases leaves unanswered the question of how much correlation there is between the proposed unsupervised metrics and existing metrics like the ROUGE score, even if they seem intuitively correlated.

- This model suffers from the usual symptoms of other abstractive summarization models (fluency errors, factual inaccuracies). But this shouldn't overshadow the bigger contributions of this paper, since dealing with these specific issues is still an open research problem.

---

> ### Author Response · Authors · 2018-11-13
> **Added human evaluation and further discussed limitations.**
>
> Thank you for your feedback.
>
> -We clarified in the paper that a limitation here is the summary is assumed to be in the form of a review with similar stylistic characteristics as the input reviews.
> -Although ROUGE is often used in the summarization literature, it attempts to approximate human evaluation of summaries, which is the gold standard. We have shown that the metrics we used in model development have guided us to good human evaluations.
> - Indeed the symptoms discussed in the error analysis affect all current neural text generation models, and we included them to ensure we didn't claim to have solved it.

---

### Official Review · AnonReviewer1 · 2018-11-02
**Evaluation methodology and measures are questionable and should not be adopted by the community**

**Rating:** 4
**Confidence:** 4

**Review:**

This paper proposes a method for multi-document abstractive summarization. The model has two main components, one part is an autoencoder used to help learn encoded document representations which can be used to reconstruct the original documents, and a second component for the summarization step which also aims to ensure that the summary is similar to the original document.

The biggest problem with this paper is in its evaluation methodology. I don't really know what any of the three evaluation measures are actually measuring, and there is no human subject evaluation back them up.
- Rating Accuracy seems to depend on the choice of CLF used, and at best says whether the summary conveys the same average opinion as the original reviews. This captures a small amount about the actual contents of the reviews. For example, it does not capture the distribution of opinions, or the actual contents that are conveyed.
- Word Overlap with the original documents does not seem to be a good measure of quality for abstractive systems, as there could easily be abstractive summaries with low overlap that are nevertheless very good exactly because they aggregate information and generalize. It is certainly not appropriate to use to compare between extractive and abstractive systems.
-There are many well-known problems with using log likelihood as a measure of fluency and grammaticality, such as biases around length, and frequency of the words.
It also seems that these evaluation measures would interact with the length of the summary being evaluated in ways which systems could game.

Other points:
- Multi-Lead-1: The lead baseline works very well in single-document news summarization. Since this model is being applied in a multi-document setting to something that is not news, it is hard to see how this baseline is justified.

- Despite the fact that the model is only applied to product reviews, and there seem to be modelling decisions tailored to this domain, the paper title does not specify so, which in my opinion is a type of over-claiming.

Having a paper with poor evaluation measure may set a precedent that causes damage to an entire line of research. For this reason, I am not comfortable with recommending an accept.


---
Thank you for responding to my comments and updating the paper. I have slightly raised my score to reflect this effort.

There are new claims in the results section that do not seem to be warranted given the human evaluation. The claim is that the human evaluation results validate the use of the automatic metrics. The new human evaluation results show that the proposed abstractive model performs on par with the extractive model in terms of conveying the overall sentiment and information (Table 2), whereas it substantially outperforms the extractive model on the automatic measures (Table 1). This seems to be evidence that the automatic measures do not correlate with human judgments, and should not be used as evaluation measures.

I am also glad that the title was changed to reflect the scope of the experiments. I would now suggest comparing against previous work in opinion summarization which do not assume gold-standard summaries for training. Here are two representative papers:

Ganesan et al. Opinosis: A Graph-Based Approach to Abstractive Summarization of Highly Redundant Opinions. COLING 2010.
Carenini et al. Multi-Document Summarization of Evaluative Text. Computational Intellgience 2012.

---

> ### Author Response · Authors · 2018-11-13
> **Added human evaluation (that corresponds to our metrics)**
>
> Thank you for your feedback, which we have incorporated, and resulting in, we believe, a much stronger paper. We agree the lack of human evaluation to validate our methods was a glaring omission in the original paper. As a result, we added Table 2 with human evaluation results of the summaries on multiple dimensions, showing our model is competitive with the extractive baseline with respect to representing the overall sentiment and information in the input reviews and also the fluency. We clarified that the automatic metrics (which rank order the methods similarly) are useful for guiding model development, but the only gold standard here is human evaluation.
>
> Regarding the points about the metrics:
> Rating accuracy: The sentiment of a good summary should be reflective of the overall sentiment of the reviews. We approximate this overall segment by the average rating. We clarify that this captures a necessary aspect of the summary, but by itself is not sufficient, which is why we have other things we look at, including now human evaluation.   The “actual contents that are conveyed” is meant to be covered by the word overlap score and one of our human eval questions.
> Word overlap: we agree with your point that abstractive systems could have lower overlap because they aggregate information and generalize. We clarified in the paper that this word overlap score is included as a sanity check: it’s possible to get a high rating accuracy while talking about something completely unrelated, and a very low word-overlap would suggest something pathological. That said, it appears to rank-order similarly as our human evaluation question.
> Negative log-likelihood: In this paper now we only use this metric to compare abstractive variants. It’s true that using it to compare to the “Concatenation” baseline (now removed) was inappropriate. The gold standard for measuring fluency would be a human evaluation which we added.
>
> Other points:
> Multi-lead 1: Having proved to be a strong baseline in other summarization tasks, we sought to create an analog of Multi-lead 1 in our multi-document setting. This proved to be a reasonably strong baseline. In any case, this is simply one of several baselines which we compare against.
> We don’t want to overclaim, and we’ve modified the title of our paper to include “of Reviews” and added clarification of limitations in the Conclusion.

---

> ### Author Response · Authors · 2018-11-29
> **response to Nov 24 review modification**
>
> We want to clarify that the automatic metrics are used to guide model development. However, the final evaluation is done with humans. In future comparisons with our method, we expect a human evaluation to be done as well.
>
> We did not focus on comparing to review-specific algorithms because our method does not rely on any review-specific properties, domain-knowledge, or highly engineered features, unlike the above papers. Review-specific choices could improve the algorithm, but would take away from the generalizability of the proposed architecture/model which is our goal. The spirit of this conference is learning generic representations of data that can be widely applicable, and not to be focused on domain-specific feature engineering. We thus focused our comparison on generic approaches without heavy feature engineering. We expect follow-up work to use our model architecture on other domains with minimal changes.
>
> P.S. Just a point of clarification, despite the title, the authors of Ganesan et al. (Opinosis), say in the paper their method is “word-level extractive summarization” and not actually abstractive.

---

### Official Review · AnonReviewer2 · 2018-11-03
**Promising unsupervised approach, but clarity issues**

**Rating:** 5
**Confidence:** 4

**Review:**

Overall and positives:

The paper investigates the problem of multidocument summarization
without paired documents to summary data, thus using an unsupervised
approach. The main model is constructed using a pair of locked
autoencoders and decoders. The model is trained to optimize the
combination of 1. Loss between reconstructions of the original reviews
(from the encoded reviews) and original the reviews, 2. And the
average similarity of the encoded version of the docs with the encoded
representation of the summary, generated from the mean representation
of the given documents.

By comparing with a few simple baseline models, the authors were able
to demonstrate the potential of the design against several naive
approaches (on real datasets, YELP and AMAZON reviews).
The necessity of several model components is demonstrated
through ablation studies. The paper is relatively well structured and
complete. The topic of the paper fits well with ICLR. The paper
provides decent technical contributions with some novel ideas about
multi-doc summary learning models without a (supervised) paired
data set.

Comments / Issues

[ issue 6 is most important ]

1.  Problem presentation. The problem was not properly introduced and
elaborated. In fact, there is not a formal and mathematical
introduction of the problem, input, output, dataset and model
parameters. The notations used are not very clearly defined and are
quite handwavy, (e.g. what is V, dimensions of inputs x_i was not
mentioned until much later in the paper). The authors should make
these more precise. Similar problem with presentations of the models,
parameters, and hyperparameters.

3.  How does non-equal weighted linear combinations of l_rec and l_sim
change the results? Other variation of the overall loss function? How
do we see the loss function interaction in the training, validation
and test data? With the proposed model, these could be interesting to
observe.

4.  In equation two, the decoder seems to be very directly affecting
the quality of the output summary. Teacher forcing was used to train
the decoder in part (1) of the model, but without ground truth, I
would expect more discussions and experiments on how the Gumbel
softmax trick affect or help the performance of the output.

5.  Baseline models and metrics

(1) There should be more details on how the language model is trained,
some examples, and how the reviews are generated from the language
model as a base model (in supplement?).

(2). It is difficult to get a sense of how these metrics corresponds
to the actual perceived quality of the summary from the
presentation. (see next)

(3). It will be more relevant to evaluate the proposed design
vs. other neural models, and/or more tested and proved methods.

6. The rating classifier (CLF) is intriguing, but it's not clearly
explained and its effect on the evaluation of the performance is not
clear: One of the key metrics used in the evaluation relies on the
output rating of a classifier, CLF, that predicts reader ratings on
reviews (eg on YELP).  The classifier is said to have 72%
accuracy. First, the accuracy is not clearly defined, and the details
of the classifier and its training is not explained (what features are
its input, is the output ordinal regression).  Equation 4 is not
explained clearly: what does 'comparing' in 'by comparing the
predicted rating given the summary rating..' mean?  The classifier may
have good performance, but it's unclear how this accuracy should
affect the results of the model comparisons.

The CLF is used to evaluate the rating of output
reviews from various models. There is no justification these outputs
are in the same space or generally the same type of document with the
training sample (assuming real Yelp reviews).  That is probably
particularly true for concatenation of the reviews, and the CLF classifier
scores the concatenation very high (or  eq 4 somehow leads to highest value
for the concatenation of reviews )... It's not clear whether such a classifier is
beneficial in this context.

7. Summary vs Reviews. It seems that the model is built on an implicit
assumption that the output summary of the multi-doc should be
sufficiently similar with the individual input docs.  This may be not
true in many cases, which affects whether the approach generalizes.
Doc inputs could be covering different aspects of the review subject
(heterogeneity among the input docs, including topics, sentiment etc),
or they could have very different writing styles or length compared to
a summary.  The evaluation metrics may not work well in such
scenarios.  Maybe some pre-classification or clustering of the inputs,
and then doing summarization for each, would  help?  In the conclusions section, the
authors do mention summarizing negative and positive reviews
separately.

---

> ### Author Response · Authors · 2018-11-13
> **Added human evaluation; additions to improve clarity of paper**
>
> Thank you for your review and the very helpful, comprehensive feedback, which we strove to address. We agree that the biggest previous issue was uncertainty around the evaluation. As a result we added results of a human evaluation that directly assesses various aspects of summarization quality and showing similar results as our proxy metrics. Regarding your specific points:
>
> 1. We’ve added a more formal, mathematical description of the problem setup. Overall, we’ve tried to be clearer and and consistent with our notation.
>
> 3. Good question. We did try a larger weight on l_sim (as intuitively, this loss helps the model produce outputs that actually summarize the original review), but we did not find meaningful differences and pointed this out in the paper.
>
> 4. Although there are no ground-truth summaries for Equation 2, there are ground truth reconstructions in Equation 1. As shown in the ablations, it is crucial that the decoders are tied in this architecture. In one of our experiments, the “Early cosine loss” (also shown schematically in Appendix A), we did not need to use the Gumbel-softmax estimator and simply decoded auto-regressively from the mean vector. That experiment shows that the decoding the summary as part of training significantly improves results.
>
> 5. (1) We’ve added details about how the language model was trained in the Experimental Setup section, as well as how the reviews were generated using the language model in the “No Training” baseline in the Baselines section. The reviews in the “No training” model are generated in the same fashion as the proposed model. The purpose of this baseline is to show that optimizing the proposed loss improves the output over simply using pre-trained language models.
>
> (2) Great point. We’ve added human evaluation experiments on Mechanical Turk regarding the quality of the summaries to assess the validity of our metrics. Briefly, the results show that our metrics guided us to a good model -- the extractive and abstractive models obtain comparable results on how well they summarize information and sentiment, and the abstractive summaries are similarly fluent.
>
> (3) There are no known neural, end-to-end models for this problem setup and this being the first is one of our main contributions. We hoped that reasonable model variations and ablations would probe into the efficacy of various aspects of our model.
>
> 6. We’ve modified the Rating accuracy description to hopefully make clear that the classifier is trained by taking as input a review x (sequence of tokens) and producing probabilities over the 5 possible ratings (i.e. it is a classification problem and not a regression problem). There are no hand-engineered features. The rating with the highest probability is the predicted rating. This is then compared to the average rating of the original reviews (rounded to the nearest 1-5 star rating).
>
> In general, we agree that the classifier baseline should be applied carefully. We’ve removed the concatenation baseline because we believe it’s outside the input space of the classifier. However, we believe the rating accuracy still applies to the other models and is a useful metric. For instance, the summaries produced by our model are constrained to the review space due to the tying of the decoders. Our human evaluation experiments also agree with the trends provided by our rating accuracy metric.
>
> 7. The assumption we make is the summary should be in some sense the “centroid” of the documents it is summarizing. If there are some positive reviews, but they are mostly negative, the summary will be mostly negative which is representative. If a priori we have a notion of review importance, we could weight some reviews higher in Equation (3) rather than equally. Or as you suggest we could summarize different clusters; in this case the most natural clustering is by review rating. In Figure 3, we show how multiple reviews could be generated for the same business, but pre-clustered by rating. We also clarify that our model architecture produces summaries in the form of a single review.

---

### Public Comment · (anonymous) · 2018-10-11
**Further research about factual inaccuracies**

In Section 5.5, you mention that "Factual
accuracy is an ongoing area of research in the summarization field". Do you have some recent papers in mind that analyze or address this specific issue in generated summaries?

By the way, this is a very interesting and thought-provoking submission!

---

### Author Response · Authors · 2018-11-13
**Summary of changes, Nov 12, 2018.**

We thank all 3 reviewers for their feedback, which we have used to improve the paper. We summarize the changes here and replied individually as well to each reviewer below.

While the proxy metrics defined are useful in model development, the gold standard for evaluating summaries, human evaluation, was missing and we have now added it to validate our model. These results agree (rank order) with the automatic metrics and show that the abstractive model has comparable sentiment agreement, information agreement, and fluency with the extractive method.

We made many changes to clarify the problem more formally and described the models in more detail. We also clarified limitations of the model and metrics.

While we believe the architecture proposed can be applied to data other than reviews, we added “... of Reviews” to the title since we only showed results on reviews.

---

### Author Response · Authors · 2018-11-29
**We avoid review-specific features in our model to be more generally applicable**

Regarding usefulness/practicality of abstractive summarization, we believe the most natural form of summary for humans is language, i.e. sentences/paragraphs. Certainly extracting common bi-grams could be done, is straightforward, and has been done in review-specific summarization systems in prior work, but is in our opinion less natural. That is a different problem than what we’re trying to solve.

Regarding the comments for comparisons with opinion-based summarization models, we also sought to create a general, domain-agnostic model architecture that could be applied to non-review documents by not relying on any review-specific features.

---

### Meta-Review · Area_Chair1 · 2018-12-14
**Interesting work but not mature enough**

**Confidence:** 4
**Recommendation:** Reject

**Metareview:**

This paper introduces a method for unsupervised abstractive summarization of reviews.

Strengths:

(1) The direction (developing unsupervised multi-document summarization systems) is exciting

(2) There are interesting aspects to the model

Weaknesses:

(1)  The authors are clearly undecided how to position this work: either as introducing a generic document summarization framework or as an approach specific to summarization of reviews. If this is the former, the underlying assumptions, e.g., that the summary looks like a single document in a group is problematic. If this is the latter, then comparison to some more specialized methods are lacking (see comments of R1).

(2) Evaluation, though improved since the first submitted version (when human evaluation was added), is still not great (see R1 / R3). The automatic metrics are not very convincing and do not seem to be very consistent with the results of human eval. I believe that instead or along with human eval, the authors should create human written summaries and evaluate against them. It has been done for extractive multi-document summarization and can be done here. Without this, it would be impossible to compare to this submission in the future work.

(3) It is not very clear that generating abstractive summaries of the form proposed in the paper is an effective way to summarize documents.  Basically, a good summary should reflect diversity of the opinions rather than reflect an average / most frequent opinion from tin the review collection.  By generating the summary from a review LM, the authors make sure that there is no redundancy (e.g., alternative views) or contradictions. That's not really what one would want from a summary  (See R3 and also non-public discussion with R1)

Overall, I'd definitely like to see this work published but my take is that it is not ready yet.

R1 and R2 are relatively negative and generally in agreement.  R3 is very positive. I share excitement about the research direction with R3 but I believe that concerns of R1 and R2 are valid and need to be addressed before the paper gets published.